# Impact of Virtual Clinics on Diabetes Distress and HbA1c Levels Among Patients with Diabetes Mellitus in Saudi Arabia

**DOI:** 10.3390/medicina61020234

**Published:** 2025-01-28

**Authors:** Mohammed A. Almarzooq, Hussain A. Almarzoug, Mohammed Jassim Alhassan, Mukhtar Ibrahim Alrashed, Jawad S. Alnajjar, Noor Abdullah Albejais, Suha Albahrani, Ibrahim A. Alibrahim, Abdullah Almaqhawi

**Affiliations:** 1College of Medicine, King Faisal University, Al-Ahsa 31982, Saudi Arabia; mohammed.almarzooq71@gmail.com (M.A.A.); hussain_2022@hotmail.com (H.A.A.); mohd9jassim@gmail.com (M.J.A.); mukthar.alrashed@gmail.com (M.I.A.); 2General Practitioner, Al-Ahsa Health Cluster, Al-Ahsa 31982, Saudi Arabia; noor.albejais@gmail.com; 3Department of Family and Community Medicine, College of Medicine, King Faisal University, Al-Ahsa 31982, Saudi Arabia; sualbahrani@kfu.edu.sa (S.A.); aalmuqahwi@kfu.edu.sa (A.A.); 4Endocrine Consultant, Endocrine and Diabetes Center, King Fahad Hospital Al Hofuf, Al Hofuf 36441, Saudi Arabia; ibrahimai3@moh.gov.sa

**Keywords:** diabetes mellitus, diabetes distress scale, virtual clinics, telemedicine, chronic disease management, patient stress, diabetes care

## Abstract

*Background and Objectives*: Diabetes mellitus is a prevalent chronic disease caused by inadequate insulin secretion or ineffective insulin response, leading to complications such as retinopathy, nephropathy, heart attacks, and strokes. Recently, “diabetes distress (DD)” has emerged as a concept, highlighting the significant emotional burden of managing diabetes, which can impact disease outcomes. Thus, this study evaluates the impact of virtual clinics on diabetes distress and glycemic measures in individuals with diabetes mellitus. *Materials and Methods*: A cross-sectional study was conducted between May and August 2024 at the Endocrine and Diabetes Center in Alahsa, Saudi Arabia, targeting persons aged 18 and older with diabetes who had engaged in-person clinics, virtual clinics, or both between 2019 and 2024. Data were collected through structured phone interviews, supplemented by laboratory results from clinical records. The survey included demographic details, diabetes information, and the Diabetes Distress Scale. Statistical analyses, including descriptive statistics, were performed to explore the relationships between diabetes distress, clinic visit type, and glycemic control, with Mann–Whitney and Chi-Squared tests used to compare variables between two groups. *Results***:** Of the 108 participants, 55.6% were male, with a mean age of 38.5 years. Type 2 diabetes was reported in 51.9% of individuals, while 48.1% had type 1. High emotional burden (44.4%) and regimen-related distress (28.7%) were prevalent, particularly among individuals with suboptimal glycemic control. While virtual visits were not significantly correlated with lower distress levels, individuals with suboptimal glycemic control exhibited significantly higher diabetes distress across various domains, including emotional and regimen-related distress (*p* < 0.05). Laboratory analysis showed a median HbA1c of 8.2%, with poor control associated with greater distress. *Conclusions*: Diabetic individuals with suboptimal glycemic control report higher diabetes distress levels, underscoring the need for integrated psychological support in DM care. Although virtual clinic visits did not significantly reduce distress, they provide a feasible option for individual follow-up.

## 1. Introduction

Diabetes mellitus (DM) is a chronic disease with a high prevalence worldwide. Diabetes mellitus (DM) is a chronic condition that affects a significant portion of the global population, with the International Diabetes Federation estimating that approximately 10.5% of the global population has diabetes, with an even higher rate of 17.7% in Saudi Arabia [1]. DM is classified into two main types. Type 1 diabetes involves the destruction of β-cells, resulting in an absolute deficiency of insulin primarily driven by autoimmune processes. On the other hand, type 2 diabetes is characterized by a spectrum ranging from predominant insulin resistance with relative insulin deficiency to a primary defect in insulin secretion combined with insulin resistance [2]. Insulin plays a crucial role in regulating growth and the metabolism of glucose, lipids, and proteins [3]. Without insulin, individuals with DM may experience persistent hyperglycemia, leading to complications such as microvascular issues (retinopathy, nephropathy) and macrovascular problems (heart attacks, strokes, and decreased blood flow to the legs) [4,5,6].

Recently, diabetes distress (DD)—an emotional burden that does not meet the diagnostic criteria for Major Depressive Disorder (MDD)—has gained attention in the diabetes literature [7]. It refers to significant emotional responses to the diagnosis of diabetes, the risk of complications, the challenges of self-management, and the lack of supportive social or structural environments [8]. Moreover, most DM care guidelines have primarily focused on the medical management of the disease, often overlooking the psychological needs of patients [9]. The overall prevalence of diabetes distress ranged from 22.3% to 48.5% [10,11]. Risk factors contribute to DD, including complex medication regimens, insulin use, medication adherence issues, high body mass index (BMI), comorbid depressive symptoms, and being female [9,12,13,14].

Diabetes is a chronic disease requiring regular visits to assigned physicians. However, the COVID-19 pandemic introduced challenges and increased health risks associated with in-person visits, prompting a shift to virtual healthcare. In-person visits increased health risks, prompting many health systems to transition to virtual visits [15]. Several studies have indicated that telehealth services have contributed to improved glycemic control and management of other clinical outcomes in individuals with type 2 diabetes during the pandemic [16]. Furthermore, this study aims to thoroughly evaluate diabetes distress and glycemic measures among individuals with diabetes, specifically focusing on the role of virtual clinics. By understanding how virtual healthcare impacts diabetes distress, we can identify key factors that influence the well-being of individuals with diabetes and improve overall diabetes management.

## 2. Materials and Methods

### 2.1. Study Design and Setting

A cross-sectional study was conducted from May 2024 to August 2024 among individuals with diabetes mellitus at the Endocrine and Diabetes Center in Alahsa, Saudi Arabia. The inclusion criteria included individuals aged 18 years or older who had attended in-person, virtual clinics, or both between 2019 and 2024. The virtual clinics comprised three key interventions: a general diabetes clinic focused on monitoring and follow-up, dietitian-led consultations providing individualized dietary guidance, and educational sessions aimed at enhancing diabetes awareness. A total of 108 participants were recruited using a convenient sampling technique to ensure representativeness. Exclusion criteria included individuals who did not have complete information or lab findings during their virtual clinic visits. Data collection was performed through structured 15–20 min phone interviews.

### 2.2. Data Collection

Data for this cross-sectional survey were collected through 15–20 min structured phone interviews [17]. For participants who were illiterate, relatives assisted in completing the survey. The survey consisted of three sections: demographic information, diabetes-related data, and the Diabetes Distress Scale [18]. Additional laboratory results, including Hemoglobin A1C (HbA1c)—categorized as less than 5.7% for normal, between 5.7% and 6.5% for prediabetes, and 6.5% or higher for diabetes, with a glycemic goal for adult diabetes control being less than 7%, fasting glucose, random glucose, Blood Urea Nitrogen (BUN), Creatinine, alanine transaminase (ALT), aspartate aminotransferase (AST), total bilirubin (3–22 umol/L), urine albumin creatinine ratio (>3.5 for nephrotic), microalbumin urine (0–30), urine creatinine (5300–22,100 umol/L), cholesterol (less than 5.2 mmol/L), HDL (more than 1.3 mmol/L), LDL (less than 3.5 mmol/L), triglycerides (less than 1.71 mmol/L), total serum protein, and serum albumin—were obtained from clinical records.

### 2.3. Statistical Analysis

The data were collected, reviewed, and then fed to Statistical Package for Social Sciences version 26 (Released 2019. Armonk, NY, USA: IBM Corp). All statistical methods used were two-tailed with an alpha level of 0.05, considering significance if the *p* value was less than or equal to 0.05. Descriptive analysis for categorical data were performed using frequencies and percentages, whereas numerical data were presented as a mean with standard deviation or a median with range for skewed data (laboratory investigations). Also, participants’ biodemographic data, diabetes data, and their medications were tabulated. The diabetes distress level for different domains were graphed besides the distribution of virtual and in-person clinic visits. Cross tabulation to show the distribution of diabetes distress among individuals with diabetes by their virtual and in-person clinic visits was performed using the Mann–Whitney test to assess the relation between diabetes control and diabetes distress level; Pearson’s Chi-Squared test or exact probability test was used for small frequency distributions. A scatter diagram with correlation analysis was used to assess the relation between visit numbers and diabetes distress score.

### 2.4. Ethical Statement

The confidentiality of all participants was ensured. Ethical approval was obtained from the Alahsa Health Cluster’s Institutional Review Board (H-05-HS-065). The study was explained to potential respondents, and informed consent was obtained from all participants. The study adhered to the tenets of the Declaration of Helsinki.

## 3. Results

A total of 108 eligible participants with diabetes were included, with their ages ranging from 18 to 65 years, with a mean age of 38.5 ± 11.6 years old. Exactly 60 (55.6%) participants were males, 66 (61.1%) were married, 31 (42.5%) had five or more children, while only 13 (17.8%) had no children. As for education, 33 (30.6%) had a bachelor’s degree/more, 13 (12%) had a diploma, 33 (30.6%) had a secondary level of education, and 29 (26.9%) had a lower level of education (Table 1). The table presents the demographic characteristics and types of DM among the study participants. In terms of age, the majority of participants with type 1 diabetes (44.2%, n = 23) are within the 18–25 years age group, while most participants with type 2 diabetes (66.1%, n = 37) are aged 45–65 years. This indicates that younger individuals are more likely to have type 1 diabetes, whereas older individuals are more likely to have type 2 diabetes. Regarding gender, 55.6% (n = 60) of the participants are male and 44.4% (n = 48) are female. Among those with type 1 diabetes, 46.2% (n = 24) are male and 53.8% (n = 28) are female. For type 2 diabetes, 64.3% (n = 36) are male and 35.7% (n = 20) are female. This suggests that type 2 diabetes is more prevalent among males, while type 1 diabetes has a slightly higher prevalence among females. In terms of marital status, 33.3% (n = 36) of participants are single, 61.1% (n = 66) are married, and 5.6% (n = 6) are divorced or widowed. Among those with type 1 diabetes, 55.8% (n = 29) are single, 40.4% (n = 21) are married, and 3.8% (n = 2) are divorced or widowed. For type 2 diabetes, 12.5% (n = 7) are single, 80.4% (n = 45) are married, and 7.1% (n = 4) are divorced or widowed. Looking at the number of children, 17.8% (n = 13) of participants have no children, 19.2% (n = 14) have 1–2 children, 20.5% (n = 15) have 3–4 children, and 42.5% (n = 31) have 5 or more children. For those with type 1 diabetes, 29.2% (n = 7) have no children, 29.2% (n = 7) have 1–2 children, 33.3% (n = 8) have 3–4 children, and 8.3% (n = 2) have 5 or more children. Among those with type 2 diabetes, 12.2% (n = 6) have no children, 14.3% (n = 7) have 1–2 children, 14.3% (n = 7) have 3–4 children, and 59.2% (n = 29) have 5 or more children. In terms of educational level, 26.9% (n = 29) have below secondary education, 30.6% (n = 33) have secondary school education, 12.0% (n = 13) have a diploma, and 30.6% (n = 33) have a bachelor’s degree. Among participants with type 1 diabetes, 21.2% (n = 11) have below secondary education, 28.8% (n = 15) have secondary school education, 7.7% (n = 4) have a diploma, and 42.3% (n = 22) have a bachelor’s degree. For participants with type 2 diabetes, 32.1% (n = 18) have below secondary education, 32.1% (n = 18) have secondary school education, 16.1% (n = 9) have a diploma, and 19.6% (n = 11) have a bachelor’s degree.

Diabetes data and medications received by study participants are presented in Table 2. A total of 56 (51.9%) had type 2 DM, 52 (48.1%) had type I. As for duration, it was less than 10 years among 36 (33.3%) cases and more than 20 years among 24 (22.2%). For treatment, 60 (55.6%) cases were on 2 types of treatment, 21 (19.4%) were on 3 types, and 20 (18.5%) were on more than 3 types. Exactly 48 (44.4%) were on weekly needles, 25 (23.1%) were on pills + insulin needles, and 14 (13%) were on pills only. The most received medications included Long-Acting insulin (69.8%), Rapid-Acting insulin (67%), Biguanides (44.3%), SGLT-2 Inhibitors (22.6%), DPP-4 Inhibitors (19.8%), Sulfonylurea (18.9%), GLP-1(12.3%), and Mixed Rapid and Intermediate insulin (1.9%).

Figure 1 highlights the distribution of the study of diabetic individuals’ virtual and in-person clinic visits in the Eastern region. The virtual clinic visits range from 0 to 13 with a mean of 3.6 ± 2.9 visits and a median of 3 visits. As for in-person visits, they range from 0 to 42 with a mean of 9.6 ± 5.9 visits and a median of 10 visits.

Table 3 illustrates the laboratory findings from the study of diabetic individuals in the Eastern region of Saudi Arabia. As for blood glucose levels, fasting glucose has a mean of 10.5 ± 5.5 with a median of 9.2. Random glucose has a mean of 10.3 ± 5.2, with the median also at 9.2. HbA1c has a mean of 8.4 ± 2.0% and a median of 8.2%. Considering renal tests, BUN has a mean of 5.7 ± 3.4, with the median being slightly lower at 5.0. Creatinine shows a wide range (mean = 84.9 ± 90.5) and a median of 65.4. The urine albumin creatinine ratio has a very high mean (49.2 ± 108.0) with a median of 5.3. Microalbumin urine and urine creatinine exhibit extreme values, with means of 248.7 and 10,874.9, respectively. For liver tests, ALT has a mean of 25.5 ± 15.6 and a median of 23.5. AST’s mean is 21.5 ± 11.6 with a median of 20.0. The total bilirubin mean is 11.5 ±5.5 with a median of 10.3. The total protein serum and albumin serum show means of 74.4 ± 5.9 and 68.6 ± 95.0, respectively. As for lipid profile, cholesterol’s mean is 4.8 ± 1.2 with a median of 4.7. HDL and LDL levels show means of 1.33 ± 0.45 and 2.85 ± 1.07, respectively. Triglycerides (TG) have a mean of 1.55 ± 0.95 and a median of 1.36.

Figure 2 demonstrates the level of diabetes distress among diabetic individuals in the Eastern region of Saudi Arabia. A total of 48 (44.4%) participants had high emotional burden, 31 (28.7%) had high regimen-related distress, 20 (18.5%) had high interpersonal distress, and 19 (17.6%) had high physician-related distress. In total, 29 (26.9%) participants had high diabetes distress levels.

The distribution of diabetes distress among diabetic individuals by their virtual and in-person clinic visits is exhibited in (Table 4). Virtual clinic visits were insignificantly higher among those with high emotional burden (3 vs. 2 visits; *p* = 0.386); it was also insignificantly higher among diabetic individuals with low Physician-related distress (3.6 vs. 3.7), Regimen-related distress (3.3 vs. 3.7 visits), and overall distress level (3.3 vs. 3.7 visits). In-person clinic visits showed similar results where they were insignificantly higher among those with low Emotional Burden (6.6 vs. 4.9), Physician-related distress (6.1 vs. 5.1), Regimen-related distress (6.5 vs. 4.5), and Interpersonal distress (6.1 vs. 5.2). Also, it was insignificantly higher among cases with low overall diabetes distress (6.2 vs. 4.9).

Figure 3 showing the correlation between virtual and in-person clinic visits with diabetic distress score. The correlation was insignificant and weak with virtual visits (rho = 0.04; *p* = 0.686) and with in-person visits (rho = 0.06; *p* = 550).

The relation between diabetes control and diabetes distress level is elucidated in Table 5. A total of 50.6% of cases with poor control had a high Emotional Burden compared to 31% of others with good control (*p* = 0.048). Also, 23.4% of cases with poor diabetes control had high Physician-related distress versus 3.4% of others with good control (*p* = 0.017). High Regimen-related distress was detected among 33.8% of cases with poor control compared to 13.8% of others with good control (*p* = 0.042). A total of 31.2% of cases with poor control had high overall diabetes distress in comparison to 13.8% of those with good control (*p* = 0.048).

Table 6 represents the distribution of laboratory findings among diabetic individuals categorized by the frequency of virtual clinic visits. Overall, the findings show no significant differences between the groups for most laboratory tests, as most *p*-values exceed 0.05. However, there are a few instances where borderline significance was observed. For example, blood glucose levels (both fasting and random) and HbA1c showed no significant differences across groups, the mean values remained relatively similar across the different visit frequencies. Similarly, there were no significant differences in renal markers such as BUN, urine albumin–creatinine ratio, and microalbumin urine, though creatinine showed a borderline *p*-value of 0.125. Liver function tests such as ALT and AST showed no significant differences, although ALT levels tended to be higher with increased virtual visits. Total bilirubin, however, showed borderline significance (*p*-value = 0.105), with levels slightly higher in individuals attending more virtual clinic visits. Regarding lipid profiles, cholesterol levels showed borderline significance (*p*-value = 0.129), with a slight decrease as the number of visits increased, while no significant differences were observed for HDL, LDL, or triglycerides.

## 4. Discussion

Telemedicine is defined by the World Health Organization as the delivery of healthcare services, where distance is a critical factor, by all healthcare professionals using information and communication technologies for the exchange of valid information for the diagnosis, treatment, and prevention of diseases and injuries, research and evaluation, and for the continuing education of healthcare providers—all in the interest of improving the health of individuals and their communities [19]. In the field of diabetes care, telemedicine has become a valuable tool for enhancing healthcare access and clinical outcomes for diabetic individuals. With many aspects of life moving online, routine management of chronic diseases is also shifting towards virtual care [20]. This study investigates the impact of virtual clinics on diabetes-related distress and glycemic control among individuals with diabetes mellitus in Saudi Arabia.

Our findings indicate a statistically significant association between virtual clinic visits and better HbA1c control. Patients utilizing telemedicine demonstrated notable reductions in HbA1c levels for both type 2 [21,22] and type 1 [23] diabetes. Prior meta-analyses suggest that most telemedicine approaches lead to meaningful reductions in HbA1c compared to standard care, with reductions ranging between 0.37% and 0.71%. Among these approaches, teleconsultation proved to be the most effective, followed by combined telecase-management and telemonitoring, and then tele-education coupled with telecase management [24].

Aragona et al. further highlighted the impact of lifestyle stability on diabetes management, reporting that the COVID-19 pandemic’s “lockdown effect” resulted in a modest but statistically significant improvement in glycemic control among individuals with type 1 diabetes. This improvement was attributed to more consistent daily routines and reduced work-related stress [25]. Additionally, a study focusing on rural individuals with type 2 diabetes using a virtual primary care model within the Veterans Affairs system demonstrated comparable glycemic and blood pressure control to in-person care, with a greater proportion of patients meeting diabetes quality standards [26].

Our study found that there is no significant relationship between virtual clinics and the diabetic distress scale in individuals with diabetes mellitus. Diabetes distress, stemming from stress and coping challenges, significantly impacts individuals’ emotional well-being. Telemedicine programs, like the Onduo Virtual Diabetes Clinic, offer effective tools for reducing diabetes-related distress, as highlighted by William H. Polonsky and colleagues [27]. Moreover, participation in the virtual group visits program led to significant increases in diabetes knowledge and support, as well as a reduction in diabetes-related distress [15]. Diabetic individuals with suboptimal glycemic control tend to experience elevated levels of emotional burden, physician-related distress, regimen-related distress, and overall diabetes-related distress compared to those with better control. The total distress score in individuals with diabetes showed significant positive correlations with factors such as longer duration of diabetes, gender, longer intervals between visits, presence of diabetes complications, hospital admissions due to diabetes-related conditions, neuropathy, severe hypoglycemia, and dyslipidemia. However, no significant correlations were observed between distress levels and variables like age, marital status, education, income, employment, smoking, physical activity, sleep duration, treatment regimen, or the presence of conditions such as cerebrovascular accident, ischemic heart disease, peripheral vascular disease, retinopathy, nephropathy, hypertension, and obesity [28].

A theory-driven stress management intervention grounded in social cognitive theory could potentially alleviate stress, improve coping self-efficacy, enhance stress management skills, and increase perceived social support, contributing to lower HbA1c levels in diabetic individuals [29]. However, acute but short-lived mental stress—marked by elevated heart rate, blood pressure (reflecting sympathetic activation), and salivary cortisol—has minimal impact on glucose control in type 1 diabetic individuals [30]. Stress assessments are particularly relevant for individuals with type 2 diabetes, as they tend to report higher stress levels than those without diabetes. Incorporating stress assessment and counseling into diabetes management could provide a more holistic approach, potentially enhancing glycemic control, reducing stress, and improving compliance among diabetic individuals [31]. A study in a tertiary diabetes care clinic in Chennai, Tamil Nadu, India, reported that 35% of individuals with diabetes experienced high to very high stress levels. Stress was notably associated with being aged 30–40, working in professional jobs, and lack of physical activity [32].

In summary, while virtual clinics can enhance glycemic control, their impact on diabetes distress remains less clear. Future research should explore integrated approaches that address both glycemic management and the psychological needs of individuals to optimize diabetes care.

### Limitations

This study’s limitations include its cross-sectional design, which prevents the establishment of causality, and potential recall and response biases due to phone interviews and reliance on family assistance for illiterate participants. Additionally, the timing of laboratory data may not fully align with survey responses, potentially affecting the accuracy of the data. The single-center setting in Al-Ahsa, Saudi Arabia, with a limited sample size may restrict the generalizability of the findings. Additionally, the study did not consider lifestyle factors, such as diet and exercise, which could impact diabetes distress and glycemic control. Future research should focus on longitudinal studies to better understand the relationship between virtual clinic interventions, diabetes distress, and glycemic control. Exploring factors such as patient engagement, the quality of virtual consultations, and the lack of integrated psychological support could provide valuable insights and help improve outcomes.

## 5. Conclusions

This study provides valuable insights into diabetes-related distress (DD) among individuals in Saudi Arabia’s Eastern region, highlighting that emotional and regimen-related burdens are significantly more prevalent among those with suboptimal glycemic control. Notably, the high levels of distress observed in individuals requiring multiple treatments underscore the complexity of diabetes management in this population. Although there was no significant association between clinic visits and reduced distress, individuals with poor diabetes management reported higher distress levels. These findings emphasize the urgent need for integrating psychological support into diabetes care, which is often overlooked in current practice. Given the unique sociocultural context of Saudi Arabia, incorporating mental health resources tailored to address emotional and regimen-related burdens could provide a novel, holistic approach to improving diabetes outcomes in the region

## Figures and Tables

**Figure 1 medicina-61-00234-f001:**
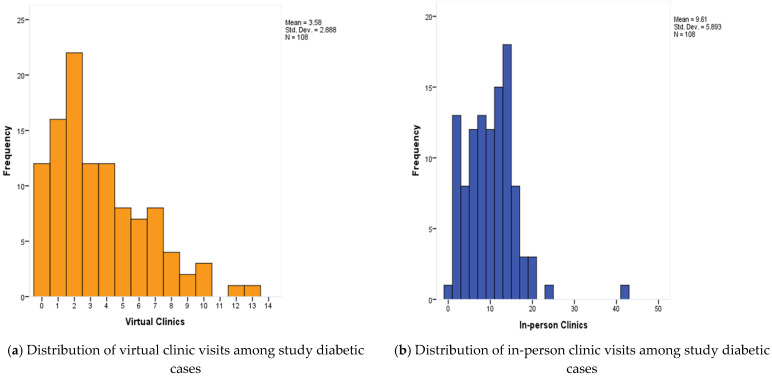
Distribution of the study diabetic individuals’ virtual and in-person clinic visits, Eastern region, Saudi Arabia.

**Figure 2 medicina-61-00234-f002:**
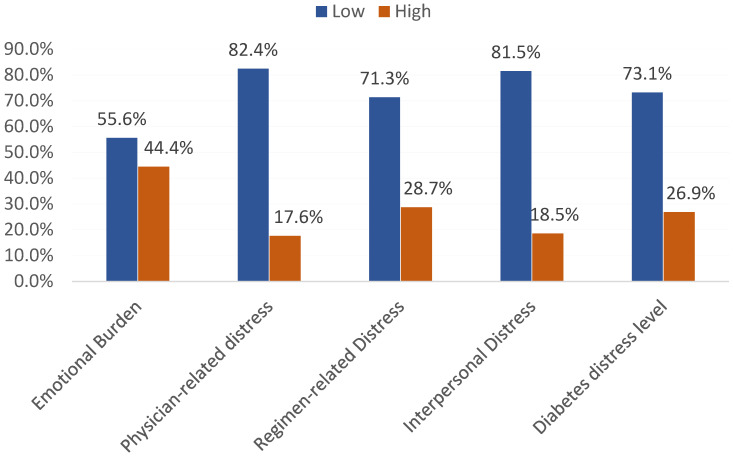
Diabetes distress level among diabetic individuals, Eastern region, Saudi Arabia.

**Figure 3 medicina-61-00234-f003:**
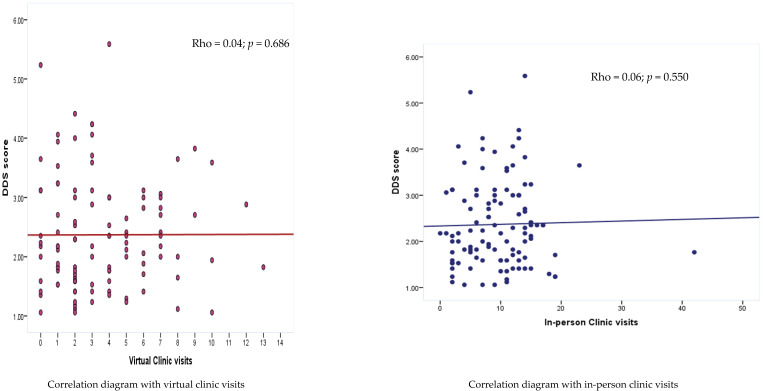
Scatter diagram showing the correlation between virtual and in-person clinic visits with diabetic distress score; rho for virtual = 0.04; *p* = 0.686; rho for in-person visits = 0.06; *p* = 550.

**Table 1 medicina-61-00234-t001:** Socio-demographic characteristics of individuals with diabetes mellitus, Eastern region, Saudi Arabia (n = 108).

Demographics	Total		Type of DM			
			Type 1 (n = 52)		Type 2 (56)	
	No	%	No	%	No	%
**Age in Years**						
18–25	24	22.2%	23	44.2%	1	1.8%
26–44	40	37.0%	22	42.3%	18	32.1%
45–65	44	40.7%	7	13.5%	37	66.1%
**Gender**						
Male	60	55.6%	24	46.2%	36	64.3%
Female	48	44.4%	28	53.8%	20	35.7%
**Marital Status**						
Single	36	33.3%	29	55.8%	7	12.5%
Married	66	61.1%	21	40.4%	45	80.4%
Divorced/widow	6	5.6%	2	3.8%	4	7.1%
**Amount of Children**						
None	13	17.8%	7	29.2%	6	12.2%
1–2	14	19.2%	7	29.2%	7	14.3%
3–4	15	20.5%	8	33.3%	7	14.3%
5 and more	31	42.5%	2	8.3%	29	59.2%
**Educational Level**						
Below secondary	29	26.9%	11	21.2%	18	32.1%
Secondary school	33	30.6%	15	28.8%	18	32.1%
Diploma	13	12.0%	4	7.7%	9	16.1%
Bachelor’s degree	33	30.6%	22	42.3%	11	19.6%

DM: Diabetes mellitus.

**Table 2 medicina-61-00234-t002:** Diabetes data and medication usage among individuals with diabetes, Eastern region, Saudi Arabia (n = 108).

Diabetes Data	No	%
**Type of DM**		
Type 1	52	48.1%
Type 2	56	51.9%
**Duration of DM in years**		
<10 years	36	33.3%
10–19 years	48	44.4%
>20 years	24	22.2%
**Number of Treatments Received**		
1	7	6.5%
2	60	55.6%
3	21	19.4%
4	11	10.2%
5	9	8.3%
**Type of Treatment Used**		
Weekly needles	48	44.4%
Pills + insulin needle	25	23.1%
Pills only	14	13.0%
Pills + weekly needles	11	10.2%
Pills + insulin needle + weekly needles	5	4.6%
Insulin needle	5	4.6%
**Medications Received**		
Long Acting	74	69.8%
Rapid Acting	71	67.0%
Biguanides	47	44.3%
SGLT-2 Inhibitors	24	22.6%
DPP-4 Inhibitors	21	19.8%
Sulfonylurea	20	18.9%
GLP-1	13	12.3%
Others	3	2.8%
Mixed Rapid and Intermediate insulin	2	1.9%

DM: Diabetes mellitus, DPP-4: Dipeptidyl Peptidase-4, SGLT-2: Sodium-Glucose Cotransporter 2, GLP-1: Glucagon-Like Peptide-1.

**Table 3 medicina-61-00234-t003:** Laboratory findings among study individuals with diabetes mellitus, Eastern region, Saudi Arabia.

Lab Investigations	Minimum	Maximum	Mean	SD	Median
**Blood glucose level**					
Fasting Glucose (mmol/L)	3.4	33.7	10.5	5.5	9.2
Glucose Random (mmol/L)	2.9	33.7	10.3	5.2	9.2
Hba1c (%)	5.1	14.3	8.4	2.0	8.2
**Renal tests**					
BUN (mmol/L)	1.6	22.3	5.7	3.4	5.0
Creatinine (umol/L)	18.1	792.8	84.9	90.5	65.4
Urine albumin creatinine ratio (mg/mmol)	0.1	444.6	49.2	108.0	5.3
Microalbumin urine (mg/L)	1.0	4395.5	248.7	787.1	7.0
Urine creatinine (umol/L)	3.2	38,114.0	10,874.9	7796.1	8967.0
**Liver tests**					
ALT (U/L)	7.0	121.0	25.5	15.6	23.5
AST (U/L)	3.0	106.0	21.5	11.6	20.0
Total bilirubin (umol/L)	3.1	41.5	11.5	5.5	10.3
Protein total serum (g/L)	54.1	97.5	74.4	5.9	75.0
Albumin serum (g/L)	28.5	469.0	68.6	95.0	43.5
**Lipid profile**					
Cholesterol (mmol/L)	2.3	8.2	4.8	1.2	4.7
HDL (mmol/L)	0.60	3.37	1.33	0.45	1.21
LDL (mmol/L)	0.09	5.86	2.85	1.07	2.75
TG (mmol/L)	0.38	5.67	1.55	0.95	1.36

Hba1c: Hemoglobin A1C, BUN: Blood Urea Nitrogen, ALT: Alanine Transaminase, AST: Aspartate Aminotransferase. HDL: High-Density Lipoprotein, LDL: Low-Density Lipoprotein, TG: Triglycerides.

**Table 4 medicina-61-00234-t004:** Distribution of diabetes distress among individuals with diabetes mellitus based on virtual and in-person clinic visits.

Distress Domain	Virtual Clinics			In-Person Clinics		
	Mean	SD	Median	Mean	SD	Median
**Emotional Burden**						
Low	3.3	2.7	2	9.4	6.6	10
High	3.9	3.1	3	9.9	4.9	10
*p*-value	0.386			0.301		
**Physician-related distress**						
Low	3.7	2.9	3	9.7	6.1	10
High	3.6	2.8	3	9.1	5.1	9
*p*-value	0.810			0.547		
**Regimen-related distress**						
Low	3.7	2.9	3	9.6	6.5	9
High	3.3	2.8	3	9.7	4.2	11
*p*-value	0.472			0.579		
**Interpersonal distress**						
Low	3.6	2.8	3	9.6	6.1	10
High	3.7	3.2	3	9.5	5.2	9
*p*-value	0.930			0.934		
**Diabetes distress level**						
Low	3.7	2.9	3	9.6	6.2	9
High	3.3	2.8	3	9.6	4.9	11
*p*-value	0.559			0.906		

*p*: Mann–Whitney test.

**Table 5 medicina-61-00234-t005:** Association between diabetes control and diabetes distress levels.

Distress Domain	Glycemic Control				*p*-Value
	Good (<7%) (n = 29)		Suboptimal (>7%) (n = 77)		
	No	%	No	%	
**Emotional Burden**					0.048 *
Low	20	69.0%	38	49.4%	
High	9	31.0%	39	50.6%	
**Physician-related distress**					0.017 *^
Low	28	96.6%	59	76.6%	
High	1	3.4%	18	23.4%	
**Regimen-related distress**					0.042 *^
Low	25	86.2%	51	66.2%	
High	4	13.8%	26	33.8%	
**Interpersonal distress**					0.169 ^
Low	26	89.7%	60	77.9%	
High	3	10.3%	17	22.1%	
**Diabetes distress level**					0.048 *
Low	25	86.2%	53	68.8%	
High	4	13.8%	24	31.2%	

*p*: Pearson X^2^ test ^: Exact probability test * *p* < 0.05 (significant).

**Table 6 medicina-61-00234-t006:** Laboratory findings distribution based on virtual clinic visits.

Labs	Virtual Clinic Times								*p*-Value
	None		1–2 Times		3–5 Times		>5 Times		
	Mean	SD	Mean	SD	Mean	SD	Mean	SD	
**Blood glucose level**									
Fasting Glucose (mmol/L)	11.52	7.49	10.37	6.27	9.44	3.50	11.64	5.67	0.472
Glucose Random (mmol/L)	8.98	3.03	10.31	5.99	9.92	4.39	11.35	5.77	0.659
Hba1c (%)	8.47	2.18	8.18	1.90	8.22	1.86	8.75	2.24	0.694
**Renal tests**									
BUN (mmol/L)	4.78	1.80	5.79	2.96	5.26	2.85	6.71	4.75	0.294
Creatinine (umol/L)	68.11	24.42	81.48	64.38	67.30	46.51	122.00	159.64	0.125
Urine albumin creatinine ratio (mg/mmol)	121.85	195.06	42.35	83.44	41.94	109.47	45.86	103.74	0.331
Microalbumin urine (mg/L)	290.09	589.98	274.85	904.05	166.97	510.35	301.28	979.89	0.983
Urine creatinine (umol/L)	10,411.82	6785.04	12,705.14	9660.93	10,743.03	7618.66	8238.14	3449.30	0.248
**Liver tests**									
ALT (U/L)	20.08	9.52	28.49	20.37	26.34	13.34	22.71	11.83	0.302
AST (U/L)	18.92	6.92	23.97	16.45	21.40	9.83	19.40	5.58	0.405
Total bilirubin (umol/L)	9.87	3.39	13.21	6.79	10.79	4.83	10.47	4.54	0.105
Protein total serum (g/L)	73.47	6.36	73.99	5.82	75.13	6.53	74.57	5.49	0.854
Albumin serum (g/L)	124.06	154.71	57.97	77.02	71.87	103.26	58.95	77.99	0.343
**Lipid profile**									
Cholesterol (mmol/L)	5.37	1.43	4.46	1.16	4.88	1.26	4.96	0.85	0.129
HDL (mmol/L)	1.47	0.58	1.30	0.47	1.35	0.44	1.30	0.38	0.780
LDL (mmol/L)	3.28	1.27	2.54	1.15	2.93	1.09	3.04	0.74	0.155
TG (mmol/L)	1.63	1.03	1.50	1.02	1.47	0.65	1.69	1.15	0.811

*p*: One Way ANOVA. Hba1c: Hemoglobin A1C, BUN: Blood Urea Nitrogen, ALT: Alanine Transaminase, AST: Aspartate Aminotransferase. HDL: High-Density Lipoprotein, LDL: Low-Density Lipoprotein, TG: Triglycerides.

## Data Availability

All relevant data supporting the findings of this study are included within the manuscript.

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
