# Peer review of "Impact of Virtual Clinics on Diabetes Distress and HbA1c Levels Among Patients with Diabetes Mellitus in Saudi Arabia"

_medicina, 2025, doi:10.3390/medicina61020234_

Round 1

Reviewer 1 Report

Comments and Suggestions for Authors

This cross-sectional study examines the potential relationship between follow-up using virtual visits and distress levels in a population of individuals with type 1 and type 2 diabetes. The hypothesis is that a greater number of virtual visits would be associated with lower distress. 

However, the results do not demonstrate any significant relationship in this regard. Nonetheless, the authors identified higher distress levels in individuals with suboptimal glucose control, which is not a novel finding.

Overall, the study appears interesting, and the methodology is acceptable. However, there are several aspects that, in my opinion, need improvement to enhance the manuscript's quality and readability for its audience. 

Below are my suggestions:

-              Abstract: In the "Methods" section, be more precise regarding statistical methods: Mann-Whitney and Chi-Square tests are used to compare variables between two groups, not to establish relationships between different variables.

-              Methods: The study period is reported as May to August 2024, while the visits occurred between 2019 and 2024, which is somewhat confusing. I recommend clarifying the study timeline.

-              Introduction: In lines 49-50, the authors should provide a more thorough definition of type 1 and type 2 diabetes.

-              Study Design and Setting: Are there exclusion criteria? The sentence on lines 82-83 is incomplete.

-              Methods Clarity: It is unclear whether each participant underwent both in-person and virtual visits or exclusively one or the other. Please clarify this point in the methods section.

-              Line 89: The word "survey" appears to be missing after "completing the."

-              Lines 91-100: I suggest rewriting this section to improve readability and logical flow.

-              Sections 2.3 and 2.5: These sections are identical. Please double-check the manuscript before submission.

-              Results: In the "Diabetes data" section and Table 2, I suggest dividing the table into two parts, one for type 1 and one for type 2 diabetes. Conducting descriptive analysis on the general population is less meaningful; stratifying by diabetes type would be more clear.

-              Table 3: Units of measurement are missing. Is the distribution of variables normal? This should be specified in the statistical methods section. Additionally, indicate mean and SD for normally distributed variables, and median and IQR for non-normal distributions.

-              Abbreviations: Include a legend for abbreviations at the foot of each table in alphabetical order.

-              Table 4: This table is confusing and not reader-friendly. I recommend reformatting it, as well as the accompanying text. It appears that distress scores are categorized as high and low, but the cut-off points used are unclear. These should be explicitly stated in both the methods and table legends.

-              Figure 3: Include p-values and R directly in the figure.

-              Results Section: The entire results section seems poorly structured and at times resembles a mere description of the tables and figures. I encourage the authors to rephrase this section to improve its flow and narrative.

-              Discussion: The recent COVID-19 pandemic was an important trial for the application of technology in medicine and diabetes care, particularly virtual visits. I recommend expanding the discussion on this topic. Additionally, some studies (suggested references: 10.1016/j.diabres.2021.108988 ; 10.1016/j.diabres.2020.108468) have described a beneficial effect of lockdown, and consequently virtual visits, on glycemic control. Including and discussing this evidence would strengthen the manuscript.

-              Throughout the manuscript, I recommend replacing terms like "patient" with more inclusive alternatives such as "individual" or "person with diabetes."

-              Replace "poor glycemic control" with less judgmental terminology, such as "suboptimal glycemic control."

Comments on the Quality of English Language

I recommend seeking the assistance of a native English speaker with expertise in scientific writing to refine the language throughout the manuscript.

Author Response

  1-Comment: Abstract: In the "Methods" section, be more precise regarding statistical methods: Mann-Whitney and Chi-Square tests are used to compare variables between two groups, not to establish relationships between different variables.

1-Response: Thank you for your feedback; we modified that in the manuscript:

 Statistical analyses, including descriptive statistics, were performed to explore the relationships between diabetes distress, clinic visit type, and glycemic control, with Mann-Whitney and Chi-Square tests used to compare variables between two groups.

2-Comment:  Methods: The study period is reported as May to August 2024, while the visits occurred between 2019 and 2024, which is somewhat confusing. I recommend clarifying the study timeline.

2-Response: Thank you. We start working on the study from May to August 2024. However, the data that were recruited from 2019 to 2024  were obtained from the system. In 2019, the hospital started the online clinics at that time.

3-Comment: Introduction: In lines 49-50, the authors should provide a more thorough definition of type 1 and type 2 diabetes.

3-Response: Thank you for your comments. We add that in the introduction.

DM is classified into two main types. Type 1 diabetes involves the destruction of β-cells, resulting in an absolute deficiency of insulin primarily driven by autoimmune processes. On the other hand, Type 2 diabetes is characterized by a spectrum ranging from predominant insulin resistance with relative insulin deficiency to a primary defect in insulin secretion combined with insulin resistance.[2] 

4-Comment: Study Design and Setting: Are there exclusion criteria? The sentence on lines 82-83 is incomplete.

4-Response: Exclusion criteria are already mentioned in line 92.

Exclusion criteria included individuals who did not have complete information or lab findings during their virtual clinic visits. Data collection was performed through structured 15–20-minute phone interviews.

5-Comment: Methods Clarity: It is unclear whether each participant underwent both in-person and virtual visits or exclusively one or the other. Please clarify this point in the methods section.

5-Response: Thanks for your feedback we have clarified this in the method section:

The inclusion criteria included individuals aged 18 years or older who had attended in-person, virtual clinics, or both between 2019 and 2024.

6-Comment: Line 89: The word "survey" appears to be missing after "completing the."

6-Response: Thanks; we modified that according to the reviewer's comments.

7-Comment: Lines 91-100: I suggest rewriting this section to improve readability and logical flow.

7-Response: Thanks; we modified that according to the reviewer's comments.

8-Comment:   Sections 2.3 and 2.5: These sections are identical. Please double-check the manuscript before submission.

8-Response: Thanks; we modified that according to the reviewer's comments.

9-Comment: Results: In the "Diabetes data" section and Table 2, I suggest dividing the table into two parts, one for type 1 and one for type 2 diabetes. Conducting descriptive analysis on the general population is less meaningful; stratifying by diabetes type would be more clear.

9-Response: Thank you for your thoughtful suggestion. While dividing the table into two parts (one for type 1 and one for type 2 diabetes) may provide additional insights, we intentionally chose not to stratify the data in this way for the following reasons:

1- We prefare to Focus on the Study Aim:

Which is to  evaluate diabetes distress (DDS) in relation to virtual vs. in-person clinic visits and Hba1c

2-Sample Size Considerations:

Splitting the data into separate tables could result in smaller subgroups, which may reduce the statistical power and clarity of the results.

We appreciate your feedback and will consider stratification in future analyses focused on diabetes type differences.

10-Comment: Table 3: Units of measurement are missing.

-Is the distribution of variables normal? This should be specified in the statistical methods section. Additionally, indicate mean and SD for normally distributed variables, and median and IQR for non-normal distributions.

10-Response: -Thanks; we ensured that the units of measurement have been provided in the table .

-Thank you for your feedback. The median was used for calculations because certain variables (Urine albumin creatinine ratio, Microalbumin urine, urine creatinine, and albumin serum) were not normally distributed. We appreciate your input and are happy to provide further clarification if needed.

11-Comment: Abbreviations: Include a legend for abbreviations at the foot of each table in alphabetical order.

11-Response: Thanks; we modified that according to the reviewer's comments.

12- Comment:   Table 4: This table is confusing and not reader-friendly. I recommend reformatting it, as well as the accompanying text. It appears that distress scores are categorized as high and low, but the cut-off points used are unclear. These should be explicitly stated in both the methods and table legends.

12-Response: Thank you for your feedback. We appreciate your suggestion regarding Table 4. The current format was chosen as it clearly aligns with the study's objectives and findings. The categorization of distress scores as high and low is based on established cut-off points detailed in the reference source ( Batais MA, Alosaimi FD, AlYahya AA, et al. Translation, cultural adaptation, and evaluation of the psychometric properties of an Arabic diabetes distress scale: A cross sectional study from Saudi Arabia. Saudi Med J. 2021;42(5):509-516. doi:10.15537/smj.2021.42.5.20200286)  and were  mentioned in method section, data collection part. We believe this approach provides the necessary clarity, but we are happy to address any specific concerns you may have.

13-Comment: Figure 3: Include p-values and R directly in the figure.

13-Response: Thanks; we modified that according to the reviewer's comments.

14-Comment:   Results Section: The entire results section seems poorly structured and at times resembles a mere description of the tables and figures. I encourage the authors to rephrase this section to improve its flow and narrative.

14-Response: Thanks; we modified that according to the reviewer's comments.

15-Comment:   Discussion: The recent COVID-19 pandemic was an important trial for the application of technology in medicine and diabetes care, particularly virtual visits. I recommend expanding the discussion on this topic. Additionally, some studies (suggested references: 10.1016/j.diabres.2021.108988 ; 10.1016/j.diabres.2020.108468) have described a beneficial effect of lockdown, and consequently virtual visits, on glycemic control. Including and discussing this evidence would strengthen the manuscript.

15-Response: Thank you very much for your valuable feedback and for providing these insightful references.

Aragona et al. further highlighted the impact of lifestyle stability on diabetes management, reporting that the COVID-19 pandemic's "lockdown effect" resulted in a modest but statistically significant improvement in glycemic control among individuals with type 1 diabetes. This improvement was attributed to more consistent daily routines and reduced work-related stress.[25]”

16- Comment:   Throughout the manuscript, I recommend replacing terms like "patient" with more inclusive alternatives such as "individual" or "person with diabetes."

16-Response: Thanks; we modified that according to the reviewer's comments.

17-Comment:  Replace "poor glycemic control" with less judgmental terminology, such as "suboptimal glycemic control."    

17-Response: Thanks; we modified that according to the reviewer's comments.

Reviewer 2 Report

Comments and Suggestions for Authors

Current report is going to demonstrate that diabetic patients with poor 37 glycemic control report higher diabetes distress levels. It is a good topic in diabetic research. However, following concerns shall be conducted.

1.       Difference between type-1 and type-2 patients was not described in clear.

2.       The diabetes distress (DD) needs to introduce in detail with reliable reference(s).

3.       Data collection was conducted via a 15-20 minute structured phone interview that needs reference(s) to support.

4.       Figure 1 needs clear legends to help.

5.       Legends in each table need to rephrase in clear. Sample size in each group was unknown.

6.       Data in Table 5 and 6 belonged to important in current study. Please revise each in clear.

7.       Conclusion of current report with novelty may strengthen it.

Author Response

1-Comment: Difference between type-1 and type-2 patients was not described in clear..

1-Response: Thank you for your thoughtful suggestion. While dividing the table into two parts (one for type 1 and one for type 2 diabetes) may provide additional insights, we intentionally chose not to 

stratify the data in this way for the following reasons:

1- We prefare to Focus on the Study Aim:

Which is to  evaluate diabetes distress (DDS) in relation to virtual vs. in-person clinic visits and Hba1c

2-Sample Size Considerations:

Splitting the data into separate tables could result in smaller subgroups, which may reduce the statistical power and clarity of the results.

We appreciate your feedback and will consider stratification in future analyses focused on diabetes type differences.

2- Comment:       The diabetes distress (DD) needs to introduce in detail with reliable reference(s).

2-Response: Thanks; we have added this modification alongside other descriptions to ensure a comprehensive understanding of diabetes distress (DD) in the manuscript :

It refers to significant emotional responses to the diagnosis of diabetes, the risk of complications, the challenges of self-management, and the lack of supportive social or structural environments. (line 62-64).

3-Comment: Data collection was conducted via a 15-20 minute structured phone interview that needs reference(s) to support.

3-Response: Thanks for valuable recommendation

We added this as a reference:

 Szolnoki, G., Hoffmann, D., Online, face-to-face and telephone surveys—Comparing different sampling methods in wineconsumer research. Wine Economics and Policy (2013), http://dx.doi.org/10.1016/j.wep.2013.10.001Batais MA, Alosaimi FD, AlYahya AA, et al. Translation, cultural adaptation, and evaluation of the psychometric properties of an Arabic diabetes distress scale: A cross sectional study from Saudi Arabia. Saudi Med J. 2021;42(5):509-516. doi:10.15537/smj.2021.42.5.20200286.

4- Comment: Figure 1 needs clear legends to help.

4-Response: Thanks; we appreciate you suggestion and have update it

 Figure 1. Distribution of the study diabetic individuals’ virtual and in-person clinic visits, Eastern region, Saudi Arabia

5- Comment: Legends in each table need to rephrase in clear. Sample size in each group was unknown.

5-Response: Thanks you for your feedback. We have rephrased the legends and for the remaining tables sample size were not mentioned, as they were not relevant to the specific data being presented.

Table 1. Socio-demographic characteristics of individuals with diabetes mellitus, Eastern region, Saudi Arabia (n=108)

Table 2. Diabetes Data and Medication Usage Among Individuals with Diabetes , Eastern region, Saudi Arabia (n=108)

Table 3. Laboratory findings among study individuals with diabetes mellitus, Eastern region, Saudi Arabia

 Table 4. Distribution of Diabetes Distress Among Individuals with Diabetes Mellitus Based on Virtual and In-Person Clinic Visits

Table 5. Association Between Diabetes Control and Diabetes Distress Levels

Table 6. Laboratory Findings Distribution Based on Virtual Clinic Visits

6- Comment:        Data in Table 5 and 6 belonged to important in current study. Please revise each in clear.   6-Response: Thank you for your helpful comment regarding Tables 5 and 6. We understand the importance of these tables in the context of the current study. In response to your suggestion, we have revised both tables for clarity, ensuring that the data is presented more clearly and is easier to interpret. Specifically, we have improved the labeling, formatting to better highlight the key findings. The revised tables are now presented in. We hope these changes address your concern and enhance the clarity of the results.   7-Comment:      Conclusion of current report with novelty may strengthen it   7-Response: Thank you for your feedback, we modified aggrading to the comments.

Reviewer 3 Report

Comments and Suggestions for Authors

The paper titled "Impact of Virtual Clinics on Diabetes Distress and HbA1c Levels Among Patients with Diabetes Mellitus in Saudi Arabia" is carefully read and reviewed. Authors evaluated the relationship between diabetes distress (DD) and glycemic control, as well as the role of virtual clinics in managing these aspects in diabetic patients in Saudi Arabia.

The authors highlighted the importance of DD, a critical yet often underappreciated factor in diabetes management. By addressing emotional and regimen-related distress, the research underscores the psychological impact of living with diabetes.

Authors effectively identified a significant association between higher DD and poor glycemic control, emphasizing the need for psychological support in diabetes care.

Several issues need to be revised and addressed.

1- Authors did not evaluate specific interventions within virtual clinics that might reduce DD, such as incorporating mental health support or patient education.

2- The near-even split between type 1 and type 2 diabetes is noteworthy but may have introduced variability in DD and glycemic outcomes. Separate analyses for these subgroups could yield more nuanced insights.

3- Authors  concluded that virtual visits did not significantly reduce DD,  however, they did not explore potential reasons, such as variations in patient engagement, the quality of virtual consultations, or the lack of integrated psychological support.

4- The study’s cross-sectional nature prevents establishing causal relationships between DD, virtual visits, and glycemic control. Longitudinal studies would be more informative. Acknowledge please.

5- The study is limited to 108 participants, which reduces the generalizability of the findings to broader populations. Elaborate as a limitation please.

Author Response

1- Comment: Authors did not evaluate specific interventions within virtual clinics that might reduce DD, such as incorporating mental health support or patient education. 1-

Response: Thanks for your feedback. We have revised the manuscript and include the following:

The virtual clinics comprised three key interventions: a general diabetes clinic focused on monitoring and follow-up, dietitian-led consultations providing individualized dietary guidance, and educational sessions aimed at enhancing diabetes awareness. 2- Comment: The near-even split between type 1 and type 2 diabetes is noteworthy but may have introduced variability in DD and glycemic outcomes. Separate analyses for these subgroups could yield more nuanced insights. 2- Response: Thank you for your thoughtful observation. While we agree that separate analyses for these subgroups could yield more specific insights, the combined analysis allows for a more comprehensive understanding of the shared experiences and challenges faced by individuals with diabetes. Future studies with larger sample sizes for each subgroup could provide a deeper exploration of the distinctions between Type 1 and Type 2 diabetes in relation to DD and glycemic outcomes 3- Comment: The authors concluded that virtual visits did not significantly reduce DD; however, they did not explore potential reasons, such as variations in patient engagement, the quality of virtual consultations, or the lack of integrated psychological support. 3- Response: Thank you for your feedback. We acknowledge this as a limitation and have included it in the limitations section, noting the need for further research on factors such as patient engagement, consultation quality, and integrated psychological support. 4- Comment: The study’s cross-sectional nature prevents establishing causal relationships between DD, virtual visits, and glycemic control. Longitudinal studies would be more informative. Acknowledge please. 4- Response: Thank you for your feedback. We acknowledge that the cross-sectional design of our study limits the ability to establish causal relationships between diabetes distress (DD), virtual visits, and glycemic control. We have addressed this in the limitation section and highlighted the need for future longitudinal studies to provide more comprehensive insights. We appreciate your valuable input. 5- Comment: The study is limited to 108 participants, which reduces the generalizability of the findings to broader populations. Elaborate as a limitation please. 5- Response: Thank you for your observation. We acknowledge that the study’s sample size of 108 participants may limit the generalizability of the findings to broader populations. This limitation has been addressed in the section, and we have highlighted the need for future studies with larger and more diverse cohorts to validate and expand upon our results. We appreciate your valuable feedback.

Round 2

Reviewer 1 Report

Comments and Suggestions for Authors

The authors have addressed all the suggestions. The quality of the manuscript has now improved.

Reviewer 2 Report

Comments and Suggestions for Authors

It has been revised in a good way. Thank you very much.